# Community engagement to increase vaccine uptake: Quasi-experimental evidence from Islamabad and Rawalpindi, Pakistan

**Mujahid Abdullah**[1], **Taimoor Ahmad**[1], **Twangar Kazmi**[1], **Faisal Sultan**[2,3], **Sabeen Afzal**[2], **Rana Muhammad Safdar**[2], **Adnan Ahmad Khan**[2,4]*

**1** Akhter Hameed Khan Foundation, Islamabad, Pakistan, **2** Ministry of National Health Services, Regulation and Coordination, Islamabad, Pakistan, **3** Shaukat Khanum Memorial Cancer Hospital & Research Centre, Lahore, Pakistan, **4** Research and Development Solutions, Islamabad, Pakistan

* adnan@resdev.org

**Data Availability Statement:** All relevant data are within the paper and its Supporting Information files.

## Abstract

Developing countries have been facing difficulties in reaching out to low-income and under-served communities for COVID-19 vaccination coverage. The rapidity of vaccine development caused a mistrust among certain subgroups of the population, and hence innovative approaches were taken to reach out to such populations. Using a sample of 1760 respondents in five low-income, informal localities of Islamabad and Rawalpindi, Pakistan, we evaluated a set of interventions involving community engagement by addressing demand and access barriers. We used multi-level mixed effects models to estimate average treatment effects across treatment areas. We found that our interventions increased COVID-19 vaccine willingness in two treatment areas that are furthest from city centers by 7.6% and 6.6% respectively, while vaccine uptake increased in one of the treatment areas by 17.1%, compared to the control area. Our results suggest that personalized information campaigns such as community mobilization help to increase COVID-19 vaccine willingness. Increasing uptake however, requires improving access to the vaccination services. Both information and access may be different for various communities and therefore a "one-size-fits-all" approach may need to be better localized. Such underserved and marginalized communities are better served if vaccination efforts are contextualized.

## Introduction

The SARS-CoV-2 (COVID-19) outbreak has been one of the largest infectious disease challenges in the past century with 559 million cases and 6.36 million deaths (as of July 15, 2022) [1]. In the initial phase, the only means to curb transmission were measures that limited contact between individuals, such as lockdowns and closures of schools, work, and public places. These, in turn, resulted in tremendous social costs and loss of well-being of individuals and societies [2]. However, the availability of effective vaccines from the first half of 2021 changed how countries and societies approached the contagion and how effective they were in doing so.

**Funding:** This work was supported, in whole or in part, by the Bill & Melinda Gates Foundation [grant number: INV-025171]. Under the grant conditions of the Foundation, a Creative Commons Attribution 4.0 Generic License has already been assigned to the Author Accepted Manuscript version that might arise from this submission. The funders had no role in study design, data collection and analysis, decision to publish, or preparation of the manuscript.

**Competing interests:** The authors have declared that no competing interests exist.

The rapidity of the vaccine development process has been unprecedented, as has been the intended scope of its coverage. Until now it took 5–10 years to develop and navigate most vaccines through regulatory approvals, which for most parts were administered to sub-sets of the population such as children, pregnant women, etc. COVID-19 vaccines went from the first identification of the virus to a public rollout of vaccines in under one year and were aimed at nearly all of the world's adult population.

This swiftness raised issues of mistrust among potential recipients, who questioned both the efficacy and safety of the vaccine [3, 4]. This in turn led to some reluctance and affected the rollout of the vaccine. Several means were attempted to promote vaccination widely, including mandates (e.g., for healthcare workers, other government personnel, or certain patients) [5, 6], tying vaccination to access to public transport or to enter stadiums, or by giving incentives to vaccinate (e.g., discounts on certain purchases). As anticipated, much of the initial vaccinations were in cities, and among well-documented and vaccine-seeking populations. However, the large-scale rollout and the intent to cover the entire population required addressing several complex situations.

For one, globally, about 1 billion people [7, 8] reside in densely populated low-income informal settlements (urban slums), where access and availability to public health facilities are limited [9]. These barriers are further accentuated by a lack of trust by public authorities that are supposed to serve them, which consider them illegal occupants of government lands (sometimes leading to forced evictions and demolitions) and often do not have a full sense of their numbers as these settlements are poorly documented [10–12]. This in turn creates social exclusions and aggravates intra-societal iniquity where the most marginalized individuals are also suspicious of the government and its initiatives to reach them with life-saving services [13–17].

Pakistan started a multi-staged rollout of COVID-19 vaccination in March 2021 that initially prioritized the oldest population, frontline workers, and those with certain risk factors, and then progressively included younger citizens, till vaccinations were opened to everyone aged 18 years or older in July 2021. When the original, voluntary uptake of vaccines slowed down, several strategies were attempted, including reaching out to poor urban communities [18] which form nearly 30% of Pakistan's total population [19]. The present study explores the effectiveness of interventions aimed at addressing demand and access barriers in such communities.

## Theoretical framework for the study

Several socio-demographic factors, communication about COVID-19 and vaccines, perceptions regarding COVID-19 vaccination, and prior experience of COVID-19 infection affect vaccine acceptance. Being male, older in age, highly educated, and employed are associated with higher acceptance; as is the perception of COVID-19 risk towards oneself and a personal or family history of COVID-19 infections [13, 20–29]. In addition, information and communication about COVID-19 vaccination act as signals to influence individual behavior [26, 27].

These factors were included in the context of the Theory of Planned Behavior (TPB) to develop a theoretical framework of vaccine willingness and uptake. The theory suggests that people's behavioral intentions are motivated by their attitudes, subjective norms, and perceived behavioral control [30]. These behavioral intentions in turn can directly affect an individual's health behavior [29–31], which we hypothesized in our study as willingness translating to increased vaccine uptake among the targeted population.

Through our community engagement interventions, we aimed to change the behavioral intentions (i.e., COVID-19 vaccine willingness) of residents in the treatment areas. Behavior

change communication was carried out in the context of social mobilization to engage the communities [32–34]. Interventions aimed at mobilizing communities for vaccination can help strengthen weak links in the causal chain, as this can enable one to take into account the local characteristics and implement the interventions more effectively [35]. When information is spread through local prominent members (community and religious leaders) of the community, people become more willing to accept it and in turn, implement it.

Engaging communities thus aid to disseminate information in the local language and channels which can have a greater outreach [33]. As found in an earlier study of rural Bangladesh [36], collaboration with non-government organizations (NGOs) to increase immunization rates also results in better service delivery and increases vaccination acceptance as people exhibit more trust in local NGOs. Mobile vaccination camps (MVCs) can help increase access to vaccinations in such underserved communities. The current study explored the roles of community mobilization and vaccination camps in marginalized, low-trust communities to promote awareness and uptake of COVID-19 vaccination and the effects of such interventions in sub-populations of these communities.

## Methodology

### Study overview, location and sampling

The study used a cross-sectional research design. Residents of five urban poor communities from the Rawalpindi-Islamabad twin cities were included. A baseline survey of 1760 respondents with equal representations of males and females was conducted from June 16 to 26, 2021, followed by an intervention (explained below). An endline survey was conducted from August 24 to September 03, 2021, with the same sampling technique, but not the same respondents. The response rate for the baseline was 98% while it was 96% for the endline.

The study was limited to COVID-19 vaccine-eligible respondents of 18 years of age that were residents of selected communities. The final survey instrument comprised 38 questions divided into multiple sections. Only a few questions were open-ended and the survey was administered in Urdu (local) language for accurate responses. Data collection was carried out in the field on electronic tablets using SurveyCTO.

Five densely populated, low-income, and underserved urban areas were selected in consultation with the Ministry of National Health Services, Regulations and Coordination (MoNHSRC) for their low participation in vaccination efforts, since such areas have historically been more hesitant towards vaccines [37–39]. Based on this experience, from Islamabad, we included I-10 (a middle-class locality), G-7 (Low-income but formal locality), F-7 (France Colony) (informal settlement), and Bhara Kahu (low- to middle-income, completely informal and recent settlement), while Dhok Hassu (low-income, long stand informal locality) was included from Rawalpindi. Each community is located sufficiently away from each other to make any cross-over contamination of the intervention (Information and eased access to local camps through social mobilization) unlikely.

Population and average household sizes were based on the Census 2017 by the Pakistan Bureau of Statistics and on-site visits (Table 1).

The sample size was calculated using MICS methodology with a 95% confidence interval [40]. We assumed a 50% acceptance rate for vaccination uptake, a design effect of 1.5, a relative margin of error of 0.12, and a 95% response rate. The sample size included 480 respondents from each of the larger communities (population greater than 30,000), while 160 each were from smaller communities.

A two-stage clustered sampling design was applied using GIS mapping, with randomization being done first by selecting a random sample of clusters in each locality of the sampling frame

**Table 1. Location characteristics of study areas.**

| Locality | Actual Population | Actual Households | Average Household Size |
|---|---|---|---|
| **Control** | | | |
| I-10 | 44,580 | 7,984 | 5.6 |
| **Treatment** | | | |
| G-7 (Low-income quarters)/F-7 (France Colony) | 38,722 | 6136 | 6.3 |
| Bhara Kahu | 125,048 | 21,123 | 5.9 |
| Dhok Hassu | 201,212 | 30,032 | 6.7 |

and then randomly selecting households from each cluster. Pins identifying clusters were dropped at random points on the map. Out of those, we randomly selected a total of 110 clusters, 30 each in Dhok Hassu, Bhara Kahu, and I-10, and 10 each in F-7 (France Colony) and G-7 (Low-income quarters) for each round of surveys.

Working in pairs, enumerators reached the pins and started with the nearest household to the left within the cluster. The pair then surveyed fifteen more households in that cluster using the left-hand rule, skipping one household after every successful interview. Each pair surveyed 8 male and 8 female respondents in every cluster, for a total of 16 interviews per cluster. Males and females were surveyed from different households, with no upper age limit restriction.

## Ethical review

The ethical review for this study was carried out by Research and Development Solutions (RADS), Islamabad which is registered with the Office for Human Research Protection (OHRP) for Institutional Review Board (IRB) approvals (Reference Number: IRB00010843). The IRB committee reviewed the research methodology, survey consent process, and survey tool, and granted a formal approval on June 8, 2021. All methods were performed in accordance with relevant guidelines and regulations. Informed written consent was taken from respondents prior to surveying given their personal information would be kept confidential and used for research purposes only.

## Interventions

**Awareness campaign via local mobilizers.** The primary intervention focused on building awareness among residents of treatment areas through social mobilization techniques geared towards improving vaccine willingness and uptake. The campaign targeted public places such as shops, markets, mosques, and churches. Within the communities, community leaders (i.e., religious and political leaders) were engaged to spread awareness of COVID-19. Printed information pamphlets (S1 and S2 Figs) were also distributed in the Urdu language via local mobilizers to explain the process of registering and getting vaccinated, while also debunking common myths surrounding vaccines, and identifying COVID-19 vaccination camps (CVCs) nearby.

**Mobile vaccination camps (MVCs).** Since there were no CVCs in the vicinity of the selected communities, MVCs were arranged to provide access to COVID-19 vaccination. These camps were organized in treatment areas in collaboration with local community-based organizations (CBOs), NGOs, and community leaders through our team of mobilizers to facilitate the community vaccination process encompassing assembling, counseling, and registering community members for the vaccination. The venue of the vaccination site was chosen by the community as a locally well-known and accessible location, such as a school or other landmarks. Local mobilizers also advertised for these in advance and on the day of the visit, they facilitated them while they were at the camp.

**Empirical measurement strategy.** We used intent-to-treat (ITT) analysis to measure average treatment effects (ATEs), assuming that households remained in the same treatment groups to which they were originally assigned, whether they received the treatments or not. We estimated ATEs on two primary outcomes: willingness to vaccinate and vaccine uptake. Given both the dependent variables were binary, we estimated non-linear ITT parameters using multi-level mixed effects logistic regressions [41] through the difference-in-differences method. Here, level 2 indicates clusters and level 1 indicates households within those clusters. As suggested by Bruhn and McKenzie [42], we did not report statistical differences between groups at baseline covariates.

The control variables in our analyses were taken based on the theoretical framework explained above as well as their predictive powers to explain the outcome variables [42], which were calculated as having strong correlations with the outcome variables. Controlling for these variables that could be imbalanced at the baseline also controlled for imbalance in the unobservable characteristics [42], and therefore the difference-in-differences analysis was applicable. We conducted all our analyses on the statistical software STATA 17.

The difference-in-differences model in regression form was then specified as follows:

$$logit(Y_{ij}) = \beta_0 + \beta_1 Treat_{ijz} + \beta_2 Time_{ijt} + \beta_3 Treat_{ijz} * Time_{ijt} + \sum_{k=4}^{20} \delta X_{ijk} + \mu_j + \varepsilon_{ij} \quad (1)$$

In Eq 1, $i$ referred to each household, $j$ referred to each cluster, z represented each treatment group, $t$ represented pre- and post-time periods and $Y_{ij}$ was the relevant outcome. Since we estimated two models, $Y_{vu}$ was vaccine uptake and $Y_{wv}$ was willingness to vaccinate. We also accounted for several control variables ($\sum_{k=4}^{20} X_{ijk}$) in our models to explain variation in our outcome variables (S1 Table). The fixed part of the model consisted of $\beta_0 + \beta_1 Treat_{ijz} + \beta_2 Time_{ijt} + \beta_3 Treat_{ijz} * Time_{ijt} + \sum_{k=4}^{20} \delta X_{ijk}$, $\mu_j$ represented the random part of the model, and $\varepsilon_{ij}$ was the household-level specific error term.

Vaccine uptake ($Y_{vu}$) was given a value of 1 if the respondent had received at least one dose of COVID-19 vaccination and 0 otherwise, and willingness to vaccinate ($Y_{wv}$) was assigned a value of 1 if the respondent was willing to get vaccinated if a free of cost government-administered COVID-19 vaccine was provided, and 0 otherwise. $Treat_{ijz}$ indicated the localities which we took in treatment and control areas. We defined three treatment areas (T1: G-7/F-7, T2: Bhara Kahu and T3: Dhok Hassu) and one control area (C: I-10). $Time_{ijt}$ was a binary variable indicating a value of 1 for post-intervention and 0 for pre-intervention time periods.

The coefficient ($\beta_3$) on $Treat_{ijz}*Time_{ijt}$ captured the effect of interventions on the treated areas as compared to the control area. Since our model was non-linear in nature, ATEs were calculated by cross derivatives with respect to $Time_{ijt}$ and $Treat_{ijz}$ variables [43]:

$$\frac{\partial^2 Y}{\partial Time\, \partial Treat} = \frac{1}{1 + e^{-(\beta_1 + \beta_2 + \beta_3 + X\beta)}} - \frac{1}{1 + e^{-(\beta_1 + X\beta)}} - \frac{1}{1 + e^{-(\beta_2 + X\beta)}} - \frac{1}{1 + e^{-(X\beta)}}$$

The R-squared for our models was calculated using a community-distributed STATA program written by Dr Wolfgang Langer [44].

## Results

At the baseline, the respondents had a median age of 35 (range: 18–86) years. They were predominantly of Punjabi ethnicity, except for Dhok Hassu, where 42% of respondents were Pashtun (Table 2). The control area (I-10) had more respondents that were Urdu speakers (10%), were better off than any of the treatment areas, and were more educated–the fewest uneducated (9%) and the most university degree holders (47%). Unemployment rates ranged from 61% in I-10 to 48% in Dhok Hassu.

**Table 2. Descriptive baseline characteristics in percentages across groups.**

| Variables | Categories | Control Group | | Treatment Groups | | | | | |
| --- | --- | --- | --- | --- | --- | --- | --- | --- | --- |
| | | I-10 | | G-7/F-7 | | Bhara Kahu | | Dhok Hassu | |
| | | Baseline | Endline | Baseline | Endline | Baseline | Endline | Baseline | Endline |
| Age Group | 18–29 | 32.2 | 28.3 | 31.7 | 36.0 | 29.2 | 26.6 | 30.8 | 29.2 |
| | 30–39 | 24.5 | 24.0 | 27.3 | 21.9 | 31.9 | 33.2 | 31.0 | 28.0 |
| | 40–49 | 19.0 | 18.7 | 19.1 | 19.3 | 22.9 | 22.7 | 22.8 | 20.9 |
| | 50–59 | 11.2 | 14.8 | 13.2 | 12.8 | 9.7 | 12.6 | 9.4 | 12.9 |
| | 60+ | 12.6 | 14.3 | 8.8 | 10.0 | 6.3 | 5.0 | 6.1 | 9.1 |
| Education level | None | 8.8 | 6.3 | 19.5 | 24.6 | 15.8 | 15.9 | 34.5 | 34.6 |
| | Up to 12 years | 44.0 | 42.3 | 57.9 | 60.1 | 63.0 | 64.5 | 58.5 | 58.1 |
| | University degree | 47.3 | 51.4 | 22.6 | 15.3 | 21.2 | 19.6 | 7.1 | 7.2 |
| Ethnicity | Punjabi | 58.0 | 59.1 | 84.0 | 90.7 | 50.6 | 63.9 | 45.1 | 48.7 |
| | Pashtun | 16.74 | 15.2 | 5.6 | 4.7 | 16.2 | 13.6 | 42.4 | 39.0 |
| | Urdu Speaking | 10.3 | 16.1 | 2.2 | 0.6 | 3.4 | 5.0 | 0.4 | 0.62 |
| | Hindko | 3.6 | 2.7 | 1.3 | 0 | 4.4 | 3.1 | 6.9 | 7.2 |
| | Others | 11.5 | 7.0 | 6.9 | 4.1 | 25.4 | 14.4 | 5.2 | 4.5 |
| Employment | Self Employed | 15.7 | 11.9 | 6.9 | 10.3 | 19.3 | 21.3 | 29.4 | 25.0 |
| | Employed | 23.1 | 22.8 | 39.8 | 38.6 | 26.9 | 22.1 | 22.4 | 21.4 |
| | Unemployed | 61.2 | 65.3 | 53.3 | 51.1 | 53.8 | 56.6 | 48.2 | 53.6 |
| Self-infection of COVID-19 | Yes | 13.5 | 13.7 | 7.3 | 4.4 | 4.3 | 3.5 | 1.5 | 2.7 |
| Family infection of COVID-19 | Yes | 16.9 | 19.7 | 11.6 | 6.0 | 7.0 | 6.4 | 2.3 | 3.3 |
| Sought treatment for last illness | Yes | 79.3 | 72.2 | 79.6 | 70.0 | 80.6 | 74.8 | 83.3 | 81.4 |
| Family vaccination | Yes | 57.4 | 85.5 | 66.9 | 89.3 | 36.4 | 69.0 | 18.9 | 57.0 |
| Distance from CVC | Up to 2 kms | 5.0 | 23.3 | 45.3 | 20.6 | 43.9 | 49.6 | 8.2 | 55.5 |
| | More than 2 kms | 49.4 | 40.3 | 42.8 | 67.6 | 29.4 | 28.1 | 37.1 | 22.3 |
| | Don't Know | 45.6 | 36.4 | 12.0 | 11.8 | 26.7 | 22.3 | 54.6 | 22.3 |
| Risk perception of COVID-19 | Worried | 65.8 | 68.5 | 63.6 | 61.7 | 59.0 | 77.9 | 71.4 | 74.1 |
| | Uncertain | 17.2 | 10.9 | 4.8 | 6.2 | 14.8 | 9.3 | 10.7 | 9.1 |
| | Unworried | 17.0 | 20.6 | 31.7 | 32.1 | 26.2 | 12.8 | 17.9 | 16.8 |
| Sources of information: | | | | | | | | | |
| Television | Yes | 26.6 | 35.0 | 33.2 | 24.6 | 25.4 | 14.2 | 23.4 | 28.3 |
| Government Call/SMS | Yes | 26.8 | 17.8 | 16.3 | 19.9 | 37.6 | 41.0 | 27.6 | 32.4 |
| Family/Friends | Yes | 46.2 | 24.2 | 49.2 | 53.0 | 46.0 | 65.2 | 44.1 | 63.1 |
| Medical professionals | Yes | 8.6 | 8.4 | 9.1 | 18.1 | 6.1 | 11.6 | 4.2 | 15.9 |
| Religious leaders | Yes | 0 | 0 | 0.3 | 7.5 | 1.5 | 2.7 | 0.4 | 6.2 |
| Any NGO/CBO working in area | Yes | 88.7 | 66.6 | 56.1 | 48.3 | 74.4 | 71.6 | 74.1 | 73.4 |

Few respondents (1% to 17%) reported any prior COVID-19 infection for themselves or among their families from any location. The highest rates were reported from the control area (I-10). However, 59–71% of all respondents reported being worried about COVID-19. The history of at least one member of the family having received the COVID-19 vaccine was the highest in G7/F7 (67%), followed by I-10 (57%), and the lowest in Dhok Hassu (19%). Distance from a CVC was the least for residents of G7/F7 and Bhara Kahu and the most for I-10 residents. Respondents from all areas reported similar proportions of sources of COVID-19 vaccination information and similar rates of treatment-seeking during a prior illness. 89% of respondents from I-10 were aware of NGOs and CBOs working in the area, compared with 56% in G-7/F-7 and 74% in Bhara Kahu and Dhok Hassu (Table 2).

**Table 3. Willingness to vaccinate and vaccine uptake segregated by gender and location.**

| | | Willingness to vaccinate | | | | | | Vaccine uptake | | | | | |
|---|---|---|---|---|---|---|---|---|---|---|---|---|---|
| | | Willing | | Uncertain | | Unwilling | | Unvaccinated | | Only registered | | At least partially vaccinated | |
| | | Baseline | Endline | Baseline | Endline | Baseline | Endline | Baseline | Endline | Baseline | Endline | Baseline | Endline |
| | **TOTAL** | 67% | 80% | 12% | 9% | 21% | 10% | 65% | 39% | 13% | 14% | 22% | 47% |
| **Male** | C: I-10 | 82% | 88% | 9% | 2% | 9% | 10% | 59% | 20% | 12% | 31% | 29% | 49% |
| | T1: G-7/F-7 | 81% | 94% | 2% | 4% | 17% | 2% | 43% | 16% | 16% | 6% | 40% | 78% |
| | T2: Bhara Kahu | 64% | 77% | 25% | 18% | 12% | 5% | 64% | 35% | 12% | 14% | 24% | 51% |
| | T3: Dhok Hassu | 64% | 85% | 13% | 10% | 23% | 5% | 76% | 38% | 14% | 22% | 10% | 40% |
| | **Total** | 72% | 85% | 13% | 9% | 15% | 6% | 62% | 28% | 13% | 19% | 24% | 52% |
| **Female** | C: I-10 | 73% | 83% | 9% | 9% | 18% | 9% | 49% | 32% | 18% | 11% | 33% | 57% |
| | T1: G-7/F-7 | 71% | 72% | 8% | 8% | 21% | 20% | 58% | 43% | 13% | 6% | 29% | 51% |
| | T2: Bhara Kahu | 51% | 80% | 13% | 4% | 36% | 16% | 79% | 60% | 8% | 7% | 12% | 32% |
| | T3: Dhok Hassu | 55% | 67% | 17% | 18% | 28% | 15% | 83% | 59% | 11% | 12% | 7% | 28% |
| | **Total** | 62% | 76% | 12% | 10% | 26% | 14% | 68% | 49% | 12% | 9% | 20% | 41% |

## Willingness to vaccinate and COVID-19 vaccine uptake

Willingness to receive vaccines increased substantially from baseline (67%) to endline (80%), more for men than women (Table 3). The control area (I-10) had the highest willingness for both men and women across both time periods, with an exception that G7/F7 had the highest willingness for men at the endline (94%).

Correspondingly, refusal to receive vaccine dropped sharply in the endline. For men, the highest dip in refusals occurred in Dhok Hassu (18%), while Bhara Kahu had the highest decrease for women (20%). Registrations and vaccinations mirror willingness. Vaccine uptake increased from 22% to 47%, with men receiving more vaccination than women (Table 3).

## Intent-to-treat (ITT) and average treatment effects (ATEs)

Average marginal treatment effects from multi-level mixed effects logistic regressions show that, compared to the control area (I-10), willingness to receive vaccination increased by 7.6% in Bhara Kahu (Coef: 0.0764, CI: 0.0121, 0.1406) and 6.6% in Dhok Hassu (Coef: 0.0661, CI: 0.00498, 0.1272) respectively. However, the change in G7/F7 was not significant.

Whereas, this willingness did not translate into an increase in vaccination rates in either of the areas. Vaccine uptake increased by 17.1% (Coef: 0.1709, CI: 0.0417, 0.3) in G-7/F-7 only (Table 4). Our adjusted models with all controls showing odds ratios are provided in S2 Table.

The Intra class correlation (ICC) is the correlation among observations within the same cluster. In our models, ICC indicates that only around 4.2–4.8% of the total variance in willingness to vaccinate and uptake is explained by between-cluster differences (i.e., due to clustering). The Mckelvey & Zavoina Pseudo R-squares of adjusted models show that 40% and 53% of the variations in willingness and uptake respectively are captured by the independent variables. Both models show good fits to predict the relevant outcomes.

## Determinants of willingness and vaccine uptake

Willingness to vaccinate was twice as likely in the control area at baseline but this effect disappeared at the endline. On the other hand, the likelihood of vaccine uptake increased in G-7/F-7 compared to the control locality (AOR: 1.975, CI: 1.079, 3.617) but not anywhere else. Women were half as likely to express willingness to vaccinate but were not any different from

**Table 4. Pairwise comparisons of average marginal treatment effects.**

| | (1) | | (2) | |
|---|---|---|---|---|
| Comparison | Willingness to vaccinate | | Vaccine Uptake | |
| | Unadjusted | Adjusted | Unadjusted | Adjusted |
| **Difference-in-differences** | | | | |
| T1: G-7/F-7 vs C: I-10 | 0.0312 | 0.0611 | 0.077 | 0.1709** |
| | (-0.0578, 0.1203) | (-0.0197, 0.1419) | (-0.0354, 0.1894) | (0.0417, 0.3) |
| T2: Bhara Kahu vs C: I-10 | 0.1189*** | 0.0764** | 0.0078 | 0.0418 |
| | (0.0424, 0.1954) | (0.0121, 0.1406) | (-0.0994, 0.115) | (-0.0817, 0.1652) |
| T3: Dhok Hassu vs C: I-10 | 0.1314*** | 0.0661** | 0.0385 | 0.0398 |
| | (0.0566, 0.2063) | (0.00498, 0.1272) | (-0.066, 0.143) | (-0.0767, 0.1563) |
| Observations | 3107 | 2,904 | 3448 | 3216 |
| Number of clusters | 220 | 220 | 220 | 220 |
| Intra-class correlation | 0.044 | 0.046 | 0.042 | 0.048 |
| McKelvey & Zavoina R2 (FE and RE) | 0.083 | 0.395 | 0.198 | 0.533 |

Robust standard errors were used, CI in parentheses

*** $p < 0.01$

** $p < 0.05$

* $p < 0.1$

men in terms of vaccine uptake. Increasing age, higher education, and employment were important determinants of willingness and uptake of vaccination at the baseline. While these factors remained important at the endline as well, their significance decreased as seen by their lowered odds at the endline. Pashtuns became less likely and Urdu speakers more likely to receive the vaccine at the endline.

Previous infection with COVID-19 for self was not a key determinant for willingness but a significant one for uptake of vaccination (AOR: 2.286, CI: 1.142, 4.576). Similarly, infection or vaccination of a family member and a high-risk perception were motivators for both willingness and uptake of vaccination. The effect of all these factors increased at the endline. Having sought treatment for a recent illness was positively correlated with uptake (AOR: 1.471, CI: 1.078, 2.009).

Having received an SMS or call from the government was a major motivator that led to increased willingness (AOR: 2.414, CI: 1.420, 4.103) and uptake (AOR: 1.310, CI: 1.004, 1.708) in the endline. Advice from friends, family, medical professionals, and religious leaders did not sway opinions about the willingness or uptake of vaccination. Living near a CVC was correlated with higher willingness compared to those who resided so far away that they did not know about the distance to a nearby CVC, and closer distances were associated with higher uptake (AOR: 4.14, CI: 2.370, 7.233 for less than 2 kms and AOR: 3.969, CI: 2.452, 6.423 for 2 + kms). The odds of vaccine uptake also increased if an NGO/CBO was working in the area in the pre-intervention period (AOR: 1.657, CI: 1.093, 2.514) (Table 5).

## Discussion

We found that interventions that raised awareness through community mobilization and removed access barriers helped improve vaccine willingness by 7% and uptake by 17% in some low-income and underserved communities. However, there is a two-stage process. In the first, awareness increased and hesitancy decreased, following our awareness interventions. In the second stage, some of those that became convinced took up the vaccines. Uptake was dependent on access to vaccinations, which our interventions addressed only in part.

**Table 5. Odds ratios of factors influencing willingness to vaccinate and vaccine uptake.**

| | Willingness to Vaccinate | | Vaccine Uptake | |
|---|---|---|---|---|
| | (1) | (2) | (3) | (4) |
| VARIABLES | Baseline | Endline | Baseline | Endline |
| **Group (Control: I-10)** | | | | |
| T1: G-7/F-7 | 0.501** | 0.766 | 0.883 | 1.975** |
| | (0.275, 0.911) | (0.365, 1.608) | (0.513, 1.520) | (1.079, 3.617) |
| T2: Bhara Kahu | 0.567** | 0.670 | 0.669 | 0.741 |
| | (0.354, 0.908) | (0.328, 1.368) | (0.405, 1.104) | (0.422, 1.301) |
| T3: Dhok Hassu | 0.718 | 1.091 | 0.483*** | 0.639 |
| | (0.435, 1.187) | (0.492, 2.418) | (0.294, 0.792) | (0.373, 1.095) |
| **Female** | 0.546*** | 0.477*** | 1.049 | 0.766 |
| | (0.349, 0.853) | (0.278, 0.818) | (0.656, 1.676) | (0.520, 1.129) |
| **Age Group (18–29)** | | | | |
| 30–39 | 1.332 | 2.072*** | 2.215*** | 1.631*** |
| | (0.924, 1.920) | (1.347, 3.187) | (1.407, 3.489) | (1.136, 2.342) |
| 40–49 | 1.699*** | 2.413*** | 6.775*** | 3.154*** |
| | (1.153, 2.502) | (1.424, 4.087) | (4.255, 10.79) | (2.135, 4.659) |
| 50–59 | 1.761** | 2.502** | 12.32*** | 4.595*** |
| | (1.010, 3.070) | (1.225, 5.111) | (7.350, 20.65) | (3.022, 6.988) |
| 60–69 | 2.913*** | 2.391** | 31.23*** | 4.890*** |
| | (1.498, 5.661) | (1.061, 5.386) | (17.22, 56.64) | (3.002, 7.966) |
| **Education level (None)** | | | | |
| Up to 12 years | 1.237 | 0.920 | 1.122 | 0.728* |
| | (0.863, 1.771) | (0.580, 1.459) | (0.695, 1.810) | (0.530, 1.001) |
| University degree | 1.941** | 0.614 | 1.735* | 0.918 |
| | (1.167, 3.227) | (0.324, 1.163) | (0.955, 3.153) | (0.576, 1.460) |
| **Ethnicity (Others)** | | | | |
| Punjabi | 0.935 | 0.996 | 1.042 | 1.054 |
| | (0.583, 1.501) | (0.483, 2.055) | (0.665, 1.635) | (0.654, 1.698) |
| Pashtun | 1.154 | 0.618 | 1.003 | 0.605* |
| | (0.667, 1.995) | (0.274, 1.393) | (0.615, 1.636) | (0.337, 1.087) |
| Urdu Speaking | 0.820 | 1.153 | 1.630 | 2.033** |
| | (0.352, 1.908) | (0.361, 3.680) | (0.808, 3.288) | (1.026, 4.027) |
| Hindko | 0.832 | 1.100 | 0.595 | 1.202 |
| | (0.357, 1.938) | (0.283, 4.269) | (0.207, 1.716) | (0.530, 2.727) |
| **Employment status (Unemployed)** | | | | |
| Self-employed | 1.339 | 2.296** | 1.009 | 1.272 |
| | (0.827, 2.166) | (1.167, 4.520) | (0.598, 1.702) | (0.834, 1.941) |
| Employed | 2.279*** | 2.396*** | 3.390*** | 2.332*** |
| | (1.412, 3.681) | (1.322, 4.339) | (2.190, 5.248) | (1.545, 3.522) |
| **Self-infection of COVID-19** | 1.079 | 0.699 | 1.095 | 2.286** |
| | (0.507, 2.299) | (0.265, 1.847) | (0.606, 1.980) | (1.142, 4.576) |
| **Family infection of COVID-19** | 3.005*** | 3.607*** | 1.877** | 1.443 |
| | (1.416, 6.376) | (1.591, 8.176) | (1.075, 3.278) | (0.840, 2.477) |
| **Family vaccination (No)** | | | | |
| Yes | 2.816*** | 4.679*** | 8.181*** | 5.766*** |
| | (1.951, 4.064) | (2.905, 7.537) | (5.175, 12.93) | (4.070, 8.169) |

(*Continued*)

**Table 5.** (Continued)

| VARIABLES | Willingness to Vaccinate | | Vaccine Uptake | |
|---|---|---|---|---|
| | (1) | (2) | (3) | (4) |
| | Baseline | Endline | Baseline | Endline |
| Not applicable | 0.296*** | 1.308 | 1.347 | 1.121 |
| | (0.168, 0.521) | (0.619, 2.767) | (0.377, 4.821) | (0.501, 2.510) |
| **Risk perception of COVID-19 (Unworried)** | | | | |
| Worried | 2.494*** | 5.627*** | 1.295 | 1.427** |
| | (1.765, 3.523) | (3.609, 8.772) | (0.833, 2.014) | (1.020, 1.995) |
| Uncertain | 0.938 | 4.021*** | 1.279 | 2.308*** |
| | (0.572, 1.540) | (1.687, 9.583) | (0.716, 2.284) | (1.247, 4.274) |
| **Source of information on COVID-19 vaccine** | | | | |
| Television | 1.107 | 1.279 | 0.823 | 1.012 |
| | (0.774, 1.585) | (0.849, 1.927) | (0.566, 1.194) | (0.752, 1.361) |
| Government Call/SMS | 1.159 | 2.414*** | 0.931 | 1.310** |
| | (0.807, 1.664) | (1.420, 4.103) | (0.637, 1.361) | (1.004, 1.708) |
| Family/friends | 1.211 | 0.889 | 0.657** | 0.774* |
| | (0.894, 1.641) | (0.559, 1.413) | (0.472, 0.916) | (0.589, 1.016) |
| Medical professionals | 2.057* | 1.046 | 1.360 | 1.187 |
| | (0.931, 4.549) | (0.544, 2.012) | (0.827, 2.236) | (0.849, 1.659) |
| Religious leaders | 0.0437*** | 2.177 | 0.131 | 1.752* |
| | (0.007, 0.265) | (0.672, 7.053) | (0.0112, 1.531) | (0.959, 3.203) |
| **Distance from CVC (Do not know)** | | | | |
| Less than 2 kms | 1.591** | 1.474 | 4.140*** | 2.273*** |
| | (1.029, 2.460) | (0.913, 2.379) | (2.370, 7.233) | (1.528, 3.381) |
| 2+ kms | 2.163*** | 2.320*** | 3.969*** | 1.765*** |
| | (1.511, 3.096) | (1.334, 4.035) | (2.452, 6.423) | (1.171, 2.661) |
| **Any NGO/CBO working in area** | 1.384* | 1.165 | 1.657** | 1.334* |
| | (0.988, 1.938) | (0.791, 1.717) | (1.093, 2.514) | (1.000, 1.780) |
| **Sought treatment for last illness** | 1.758*** | 1.470 | 0.772 | 1.471** |
| | (1.236, 2.499) | (0.882, 2.451) | (0.500, 1.192) | (1.078, 2.009) |
| **Constant** | 0.365** | 0.293* | 0.00392*** | 0.0306*** |
| | (0.144, 0.924) | (0.0852, 1.005) | (0.001, 0.0141) | (0.012, 0.078) |
| Observations | 1,417 | 1,487 | 1,613 | 1,603 |
| Number of clusters | 110 | 110 | 110 | 110 |
| Intra-class correlation | 0.038 | 0.084 | 0 | 0.087 |
| McKelvey & Zavoina R2 (FE and RE) | 0.401 | 0.434 | 0.584 | 0.423 |

Robust standard errors, CI eform in parentheses

*** p<0.01

** p<0.05

* p<0.1

Our intervention results suggest that raising awareness of COVID-19 vaccination through more personalized means at community levels using printed material in local languages, engaging with community leaders, and building partnerships with local CBOs and NGOs can improve vaccine willingness by changing the behavioral intentions of residents, which is in line with previous literature on the topic [13, 32, 35]. By contrast, merely informing the public through television, the internet or newspaper, etc., i.e., non-personalized means, may be less

effective, as was seen in the control area. However, benefits from this approach may saturate beyond a certain point. For example, our interventions were successful in improving vaccine willingness in Bhara Kahu and Dhok Hassu by 7.6 and 6.6 percentage points compared to I-10, but to a lesser extent in G7/F7 where there had already been high willingness at baseline.

While willingness improved, it did not always translate into increased uptake. A major barrier to achieving COVID-19 vaccination coverage in some settlements is the difficulty that residents have in accessing CVCs, which were often several kilometers away. While our set of interventions included some mobile vaccination centers (MVC), these were insufficient to fulfill the extent of demand for vaccination. Thus, vaccine uptake increased the most in G-7/F-7 by 17.1 percentage points compared to the control area, possibly because these areas are located in the center of Islamabad with easier access to multiple CVCs nearby–as compared to Bhara Kahu, a peri-urban slum outside Islamabad, and Dhok Hassu, an urban slum of Rawalpindi.

Table 5 tracks changes in willingness and uptake before and after the intervention. Odds of willingness to receive vaccination which was lower in all urban slums at baseline when compared to the control area became indistinguishable at the endline. Similarly, the odds of willingness for university-educated respondents, which were initially twice as much as those with lesser education, became insignificant. On the other hand, the odds of willingness rose for those that had experienced infection for self or family, if someone was already worried about infections, or if a family member had received vaccination. It appears that the interventions may have helped mitigate the disadvantage of residence in an urban slum or from lower education and accentuated the willingness of those who had encountered infection or vaccines.

Similarly, the odds of uptake of vaccination rose by two-fold in G-7/F-7, which is located in the city center, and for the 18–39 years age groups compared to all older groups. On the other hand, there appeared to be little effect from education or employment and a slight loss of advantage for those with a previous vaccination in the family. In short, in the endline period, there appears to be a homogenization effect in terms of who would take up vaccination, or a loss of disadvantage of the less educated, younger individuals, and residents of marginalized communities.

Our interventions also helped to increase vaccine willingness for people whose family members had a prior experience with COVID-19 but did not affect their uptake, which was more associated with a previous infection for self. Although our results are consistent with a prior study that uses a sample of Pakistan's adult population to the extent that an incidence of COVID-19 among family members influences perception about COVID-19 vaccination [45], the effect is not strong enough to translate this willingness into action.

Vaccine uptake was higher among those with a previous infection with COVID-19, those with risk-aversion (who were worried about getting infected), and those who sought treatment for their illness. Previous research has shown a positive association between health concerns and vaccine willingness [46–48], therefore our interventions may have nudged them to seek vaccination [49]. This implies that raising awareness, dispelling rumors, and communicating the benefits of COVID-19 vaccines can change the behavior of people who are more concerned about their health.

A prior study on routine immunizations in urban slums of Pakistan found that the source of information also plays an important role in shaping trust and risk perceptions of vaccines [39]. Given the social norms and inaccurate information especially in low-income settlements, people might not get vaccinated due to social hesitancy regarding COVID-19 vaccination. We also found that government calls and SMS about COVID-19 vaccination in the endline were associated with both increased willingness and uptake. Given the high tele-density of cellphone users in Pakistan—85.3% penetration as of October 2021 [50], cellphone campaigns communicating COVID-19 and the benefits of vaccination may be cost-effective.

## Limitations

One limitation is that a group (set) of interventions–community outreach through local leaders, local vaccination campaigns, and information–was implemented for all treatment groups. Therefore, we cannot isolate the effect of each intervention to analyze its relative effectiveness. Secondly, although MVCs were deployed in each treatment area and each reached around 150–250 vaccinations a day compared to 30–50 when they merely showed up without our intervention, such MVC visits were too few. Only 5% of the respondents out of the total vaccinated reported that they had been vaccinated through MVCs, suggesting that CVCs were the predominant source of vaccinations and subject to the distance effect mentioned above. This may have limited the impact on vaccination uptake in distant communities (Bara Kahu and Dhok Hassu). Data for previous self and family infection of COVID-19 are self-reported and not verified through laboratory means which may have skewed our results to some extent. The results of vaccine willingness may have suffered from the social desirability bias as some respondents may have provided socially acceptable responses in view of the enumerators, which may not be aligned with their actual intentions. Finally, the baseline and endline were separated by 2 months, during which the national vaccination campaign had ramped up. Some of the homogenization of uptake may be explained by this rather than an intervention effect, although the difference-in-differences analysis shows a significant change.

## Conclusion

We show that personalized information campaigns such as community mobilization and direct messaging are superior to general messaging in helping overcome COVID-19 vaccine hesitancy. However, increasing uptake of the vaccine requires an additional step of improving access to vaccination services. Both information and access may be different for various communities and therefore a "one-size-fits-all" approach may need to be better localized. These findings may apply to other vaccinations and possibly to other health initiatives where the public may require motivation to uptake services such as diabetes or hypertension screening or testing. A key lesson is that low-income or marginalized communities would be better served if the services are brought to them locally.

## Supporting information

**S1 Fig. Pamphlet distributed for awareness campaign.**
(TIF)

**S2 Fig. English translation: Pamphlet distributed for awareness campaign.**
(TIF)

**S1 Table. Names, descriptions and coding of variables.**
(PDF)

**S2 Table. Adjusted odds ratios for willingness to vaccinate and vaccine uptake.**
(PDF)

**S1 Data. The dataset used for the study.**
(DTA)

**S1 Questionnaire. The questionnaire used in the study (in English and Urdu languages).**
(PDF)

## Author Contributions

**Conceptualization:** Mujahid Abdullah, Twangar Kazmi, Faisal Sultan, Sabeen Afzal, Rana Muhammad Safdar, Adnan Ahmad Khan.

**Data curation:** Mujahid Abdullah, Twangar Kazmi.

**Formal analysis:** Mujahid Abdullah, Taimoor Ahmad.

**Funding acquisition:** Adnan Ahmad Khan.

**Investigation:** Mujahid Abdullah, Taimoor Ahmad, Twangar Kazmi, Rana Muhammad Safdar, Adnan Ahmad Khan.

**Methodology:** Mujahid Abdullah, Taimoor Ahmad.

**Project administration:** Mujahid Abdullah, Twangar Kazmi, Adnan Ahmad Khan.

**Resources:** Faisal Sultan, Sabeen Afzal, Adnan Ahmad Khan.

**Software:** Mujahid Abdullah, Taimoor Ahmad.

**Supervision:** Faisal Sultan, Sabeen Afzal, Rana Muhammad Safdar, Adnan Ahmad Khan.

**Validation:** Twangar Kazmi, Faisal Sultan, Sabeen Afzal, Rana Muhammad Safdar, Adnan Ahmad Khan.

**Writing – original draft:** Mujahid Abdullah, Taimoor Ahmad, Twangar Kazmi.

**Writing – review & editing:** Mujahid Abdullah, Taimoor Ahmad, Faisal Sultan, Sabeen Afzal, Rana Muhammad Safdar, Adnan Ahmad Khan.

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
