## [Decision Letter · Decision Letter 0]

26 Sep 2022

PONE-D-22-24493Community engagement to increase vaccine uptake: Quasi-experimental evidence from Islamabad and Rawalpindi, PakistanPLOS ONE

Dear Dr. Khan,

Thank you for submitting your manuscript to PLOS ONE. After careful consideration, we feel that it has merit but does not fully meet PLOS ONE’s publication criteria as it currently stands. Therefore, we invite you to submit a revised version of the manuscript that addresses the points raised during the review process.

We look forward to receiving your revised manuscript.

Kind regards,

Harapan Harapan, MD, PhD

Academic Editor

PLOS ONE

2. Thank you for providing the English version of the questionnaire used in this study. Please also provide a version of the questionnaire in the Urdu language.

Reviewers' comments:

Reviewer's Responses to Questions

**Comments to the Author**

1. Is the manuscript technically sound, and do the data support the conclusions?

Reviewer #1: Yes

Reviewer #2: Partly

2. Has the statistical analysis been performed appropriately and rigorously? 

Reviewer #1: Yes

Reviewer #2: Yes

3. Have the authors made all data underlying the findings in their manuscript fully available?

Reviewer #1: Yes

Reviewer #2: Yes

4. Is the manuscript presented in an intelligible fashion and written in standard English?

Reviewer #1: Yes

Reviewer #2: No

5. Review Comments to the Author

Reviewer #1: This is an excellently written study, which takes advantage of a rigorously obtained sample and robust statistics. The message is important and timely. After reading the manuscript I have only a couple of minor points:

line 58: I think "pregnant women" is more precise

line 60: I think "all the world's adult population" is more precise for the initial phase of the rollout

You mention a difference between acceptance of a vaccine and actual uptake, and explain it well through the TPB and difficulties in access. Could this also be due to some social desirability bias - of stating you were willing within a survey but then not really caring to get it?

Reviewer #2: Thank you for the reviewing opportunity. Please find below my comments on the manuscript entitled “Community engagement to increase vaccine uptake: Quasi-experimental evidence from Islamabad and Rawalpindi, Pakistan”. The study is well-design but requires several clarifications. More importantly, this study has a significance reporting the influence of distance-to-vaccination site on vaccine uptake and personalized campaign (though authors still must clarify further what personalized campaign mean in their research).

1. Please pay attention to the grammar and punctuation. Many errors were found, I suggest author to improve the English of this manuscript.

2. Line 47—48. As of?? Provide the date when the data were recorded.

3. After this, “…efficacy and safety of the vaccine. ” (Line 62), you may incorporate the following studies that reported on vaccine acceptance rate at different efficacy and safety levels:

i. Rosiello et al. Narra J 2021; 1(3): e55-doi: 10.52225/narra.v1i3.55

ii. Rayhan et al. Narra J 2022; 2(2): e85-doi: 10.52225/narra.v2i2.85

4. Line 73—77. How the data stated therein were collected? Like trust, how this was measured in the cited study? The government was shown to have negligent attitudes towards those who live in the slum area, how this was observed by the cited study? Authors have to make sure all statements are scientifically sound by checking the credibility and reliability of the data used in their statements. Please modify the paragraph.

5. “..their low participation in the ongoing vaccination efforts.” Have the data been published? Please clarify.

6. The method is already clear. But, some clarifications are needed:

I. Can you please provide the translated version for Fig 1S? It’s a good information for those who want to replicate the same approach.

II. Line 194 “…helped improve access of community…” Have been wondering what ‘access’ meant by authors. Could you please clarify?

III. Control and treatment groups are from different localities? Please indicate so in Table 1. Also, what are the rationales?

7. Line 257. “Willingness to receive vaccines increased substantially..” Where can we see the data? Table 3? Please cite the table.

8. Line 309. “Living near a CVC was correlated with higher willingness..” Based on what parameter the correlation was drawn?

9. Line 328, “engaging with community leaders, and building partnerships with local organizations” What results reflect this statement. I mean I hardly found the effect of the community leaders. And for the local organization, do you mean the NGO? Please use the same term for this issue to avoid confusion by readers.

10. Please clarify what personalized means is all about? Using local language? Or is there any other considerations for this.

11. Please comment on the context of culture, belief, and religion of the communities pertaining to the vaccine acceptance. Author may refer to this study:

Hassan et al. Narra J. 2021; 1(3): e57—doi:10.52225/narra.v1i3.57

6. PLOS authors have the option to publish the peer review history of their article (what does this mean?). If published, this will include your full peer review and any attached files.

Reviewer #1: No

Reviewer #2: No

---

## [Author Response · Author response to Decision Letter 0]

31 Oct 2022

Dear Dr Harapan,

Thank you for the invitation to revise and resubmit the manuscript titled: ‘Community engagement to increase vaccine uptake: Quasi-experimental evidence from Islamabad and Rawalpindi, Pakistan’. We also thank the reviewers for their time and feedback on this manuscript. Their collective comments have prompted us to improve the manuscript substantially. As part of our resubmission, we have attached a document named "Response to Reviewers" which includes our point-by-point response to reviewer comments. Kindly refer to that.

Warm Regards,

Adnan Khan

---

## [Decision Letter · Decision Letter 1]

14 Nov 2022

Community engagement to increase vaccine uptake: Quasi-experimental evidence from Islamabad and Rawalpindi, Pakistan

PONE-D-22-24493R1

Dear Dr. Khan,

We’re pleased to inform you that your manuscript has been judged scientifically suitable for publication and will be formally accepted for publication once it meets all outstanding technical requirements.

Kind regards,

Harapan Harapan, MD, PhD

Academic Editor

PLOS ONE

Additional Editor Comments (optional):

Reviewers' comments:

Reviewer's Responses to Questions

**Comments to the Author**

1. If the authors have adequately addressed your comments raised in a previous round of review and you feel that this manuscript is now acceptable for publication, you may indicate that here to bypass the “Comments to the Author” section, enter your conflict of interest statement in the “Confidential to Editor” section, and submit your "Accept" recommendation.

Reviewer #2: All comments have been addressed

2. Is the manuscript technically sound, and do the data support the conclusions?

Reviewer #2: Yes

3. Has the statistical analysis been performed appropriately and rigorously? 

Reviewer #2: Yes

4. Have the authors made all data underlying the findings in their manuscript fully available?

Reviewer #2: Yes

5. Is the manuscript presented in an intelligible fashion and written in standard English?

Reviewer #2: Yes

6. Review Comments to the Author

Reviewer #2: Authors have responded all of my concerns and made clarifications sufficiently. The revised version is satisfactory and can be accepted for publication. Congratulations to authors!

7. PLOS authors have the option to publish the peer review history of their article (what does this mean?). If published, this will include your full peer review and any attached files.

Reviewer #2: **No**

---

## [Editor Report · Acceptance letter]

21 Nov 2022

PONE-D-22-24493R1 

Community engagement to increase vaccine uptake: Quasi-experimental evidence from Islamabad and Rawalpindi, Pakistan 

Dear Dr. Khan:

I'm pleased to inform you that your manuscript has been deemed suitable for publication in PLOS ONE. Congratulations! Your manuscript is now with our production department. 

Kind regards, 

on behalf of

Dr. Harapan Harapan 

Academic Editor

PLOS ONE